

# Of power and despair in cetacean conservation: estimation and detection of trend in abundance with noisy and short time-series

Matthieu Authier[1,2], Anders Galatius[3], Anita Gilles[4] and Jérôme Spitz[1,5]

[1] Observatoire Pelagis UMS3462 CNRS-La Rochelle Université, La Rochelle Université, La Rochelle, France
[2] ADERA, Bordeaux, France
[3] Department of Bioscience - Marine Mammal Research, Åarhus University, Roskilde, Denmark
[4] Institute for Terrestrial and Aquatic Wildlife Research (ITAW), University of Veterinary Medicine Hannover, Foundation, Büsum, Germany
[5] Centre d'Etudes Biologiques de Chizé UMR 7372 CNRS - La Rochelle Université, CNRS, Villiers en Bois, France

Corresponding author
Matthieu Authier,
matthieu.authier@univ-lr.fr

## ABSTRACT

Many conservation instruments rely on detecting and estimating a population decline in a target species to take action. Trend estimation is difficult because of small sample size and relatively large uncertainty in abundance/density estimates of many wild populations of animals. Focusing on cetaceans, we performed a prospective analysis to estimate power, type-I, sign (type-S) and magnitude (type-M) error rates of detecting a decline in short time-series of abundance estimates with different signal-to-noise ratio. We contrasted results from both unregularized (classical) and regularized approaches. The latter allows to incorporate prior information when estimating a trend. Power to detect a statistically significant estimates was in general lower than 80%, except for large declines. The unregularized approach (status quo) had inflated type-I error rates and gave biased (either over- or under-) estimates of a trend. The regularized approach with a weakly-informative prior offered the best trade-off in terms of bias, statistical power, type-I, type-S and type-M error rates and confidence interval coverage. To facilitate timely conservation decisions, we recommend to use the regularized approach with a weakly-informative prior in the detection and estimation of trend with short and noisy time-series of abundance estimates.

## INTRODUCTION

Ecologists have long strived for power, often of the statistical kind (*Gerrodette, 1987*; *Link & Hatfield, 1990*; *Thomas, 1996*; *Seavy & Reynolds, 2007*; *White, 2018*). In particular, the issue of low statistical power to detect change in time-series of population abundance estimates arose early on (*Anganuzzi, 1993*), with obvious, and sometimes dire, consequences for applied conservation. Some twenty five years ago, *Taylor & Gerrodette (1993)* pithily warned about predicating conservation efforts on stringent statistical

requirements such as reaching the arbitrary level of 80% statistical power (associated with an arbitrary statistical significance level of 5%) to detect a decrease in abundance for the vaquita porpoise (*Phocoena sinus*), an elusive and small-bodied cetacean endemic to the Gulf of California: "if we were to wait for a statistically significant decline before instituting stronger protective measures, the vaquita would probably go extinct first (page 492)." With the vaquita now numbering less than 30 individuals, extinction is indeed imminent (*Taylor et al., 2017*; *Jaramillo-Legorreta et al., 2019*), and appears in fact unavoidable (*Parsons, 2018*). While blaming statistical power for the vaquita's quiet vanishing out of the Anthropocene would be excessive (see *Bessesen (2018)* for an overview of the vaquita case), we nevertheless think that it illustrates how ecologists may have painted themselves into a corner in their insistence for statistical 'orthodoxy' inherited from the uneasy wedding of Fisherian (statistical significance) and Neyman–Pearsonian (type-I and type-II errors) philosophies (*Hubbard & Bayarri, 2003*; *Christensen, 2005*).

Notwithstanding *Taylor & Gerrodette's (1993)* warning and changing winds in the statistical philosophies of conservationists (*Wade, 2000*; *Ellison, 2004*; *Saltz, 2011*), statistical significance and statistical power remain paramount in conservation practice. Despite widespread recognition of the need for a precautionary approach (*Trouwborst, 2009*), the burden of proof remains on the shoulders of conservationists who, in line with traditional statistical practices designed to avoid false alarms, must provide evidence of adverse effects (e.g., a decline in abundance) against an assumption of no effect (*Shrader-Frechette & McCoy, 1992*; *Noss, 1994*). High statistical power of a statistical procedure gives high confidence in results (*Buhl-Mortensen, 1996*), and may help bridge the gap between scientific uncertainty and norms of certitude for decision making (*Noss, 1994*). In the European Union, the main conservation instruments are the Habitats Directive (HD, 92/43/EEC) and the Marine Strategy Framework Directive (MSFD, 2008/56/EC) for terrestrial and marine ecosystems. In particular, Favourable Conservation Status defined by the HD requires that monitoring should be able to "detect a decline in abundance of more than 1% per year within a specific time period" (*European Commission, 2011*). MSFD set the ambitious transboundary agenda of maintaining or restoring the Good Environmental Status (GES) "of marine waters where these provide ecologically diverse and dynamic oceans and seas are clean, healthy and productive". An oft-mentioned prerequisite of GES indicators is a high statistical power to detect change over time (*Zampoukas et al., 2014*).

With respect to cetaceans, a group of species well acquainted with discussions of statistical power (*Taylor & Gerrodette, 1993*), the Olso-Paris (OSPAR) Convention for the Protection of the Marine Environment of the North-East Atlantic published in 2017 its Intermediate Assessment of GES (*OSPAR, 2017a*) and lamented on the lack of statistical power to detect change despite for example, three large scale SCANS surveys over the North-East Atlantic since 1994 (*OSPAR, 2017b*). This conclusion is hardly surprising though: some 10 years ago, *Taylor et al. (2007)* already warned of an abyssmaly low power to detect accurately precipituous decrease (defined as a 50% decline in 15 years) in the abundance of marine mammals. This result was discussed at length in subsequent expert

groups (*ICES, 2008, 2014, 2016*) yet statistical, and consequently decisional, power remained low (*ICES, 2016*; *OSPAR, 2017b*).

Three main problems with statistical power in the analysis of change in abundances of marine mammals have been identified: (i) low precision of the estimates (*ICES, 2016*), (ii) low frequency of monitoring (*ICES, 2016*), and (iii) choice of a baseline (*ICES, 2010*). All these problems boil down to the kind of data based on which a trend is to be estimated: usually noisy and short time-series. Even for the vaquita, en route to extinction, only three abundance estimates are available between 1997 and 2015, and all these estimates have coefficient of variation (CV) equal to or above 50% (*Taylor et al., 2017*). Prior to 1997, no estimate is available but the population is thought to have numbered less than 1,000 individuals (*Taylor & Gerrodette, 1993*). Although the absolute numbers of vaquita are strikingly low, the short time-series, high CVs, and imprecise baseline are typical (*Taylor et al., 2007*). These features may be intrinsic to elusive and highly mobile species such as cetaceans, but can also characterize also other species (e.g., sharks). Short time-series results from the inherent difficulties of monitoring mobile species (*Authier et al., 2017*), low precision from many uncertainties (e.g. detection probability of elusive species in heterogeneous environments; *Katsanevakis et al., 2012*) and imprecise baseline from the lack of historical quantitative data for many species (*Lotze & Worm, 2009*; *McClenachan, Ferretti & Baum, 2012*). For most marine mammals, increasing the frequencies of surveys appears as a limited option given the high costs associated with sampling large parts of the oceans. Increasing the precision of estimates can be achieved with the use of model-based estimates (such as density-surface models; *Miller et al., 2013*), at the risk of an increase in bias if the model is misspecified. There is no easy fix to increase statistical power for applied conservation in the Anthropocene.

Statistical power is equals to one minus the long-term frequency of failing to reject the null hypothesis when it is false. It is the complement of the type-II error rate in the Neyman–Pearson statistical philosophy (*Hubbard & Bayarri, 2003*). Assessing statistical power often requires Monte Carlo studies to simulate population declines in abundance and whether a proposed method can detect this decline. Such studies tends to be retrospective (*Thomas, 1997*) and, unfortunately, they are often uninformative (*Gillett, 1996*; *Thomas, 1997*; *Hoenig & Heisey, 2001*; *Lenth, 2007*) or even misleading (*Gelman & Carlin, 2014*; *Vasishth & Gelman, 2017*). The latter results from the use of statistical significant effect sizes reported in the literature: statistical significance preferentially selects badly estimated effect sizes (*Lane & Dunlap, 1978*), that can be exaggerated or even of the wrong sign (*Gelman & Tuerlinckx, 2000*; *Lu, Qiu & Deng, 2018*). Thus, the problem with statistical power may not be solely caused by measurement difficulties, but also by structural ones with Null Hypothesis Significance Testing (*Lash, 2017*). *Gelman & Tuerlinckx (2000)* introduced the concepts of type-M and type-S errors to describe the distorting consequences of statistical significance: a type-M error is an error in the magnitude of an estimated effect size given that it is statistically significant, and a type-S error is an error in the sign of an estimated effect size given that it is statistically significant. *Gelman & Tuerlinckx (2000)* further argued that type-M and type-S errors are more informative than the traditional type-I and type-II errors. To our knowledge,

methodologies currently used by ecologists and conservationists to assess and detect trends in time-series of population abundance estimates have not been investigated in terms of type-M and type-S error rates. Type-M error can be represented as an exaggeration ratio between the statistically significant estimate and its true value (when known).

Below, we perform a Monte Carlo study to investigate the statistical power, the type-M and type-S error rates of the most used technique to detect a trend in short and noisy time-series: linear regression. While this topic has been extensively covered, we provide a new outlook by focusing on pragmatic considerations, and avoiding some restrictive assumptions while making some unconventional choices. In particular, we start with general considerations on the sort of imperfect data that are available right now to conservationists. We then focus not only on detecting a trend but also on its accurate estimation, and propose to use statistical regularization with linear regression (*Gelman & Shalizi, 2013*). The latter enables to incorporate prior information and shrink estimates to address the problem of type-M errors. Our philosophical outlook is instrumentalist, rather than realist (*Sober, 1999*): we do not look for a true model, but for a wrong model that nevertheless allows correct inference on trends from sparse data, while using standard tools of modern statistical packages (e.g., the R software; *R Core Team, 2018*). Thus we investigate the frequency properties of regularized linear regression not only in terms of the traditional considerations of bias, coverage, and type-I error rates; but also with respect to type-M and type-S errors. We finally illustrate our proposed methods with real-world examples on cetacean monitoring in European waters.

## METHODS

A power analysis requires the following steps (*Lenth, 2001*):

1. a null hypothesis $\mathcal{H}_0$ on a parameter $\theta$;
2. an effect size (magnitude) of $\theta$;
3. a probability model relating $\theta$ to data (that is, a data-generating mechanism);
4. a statistical test on $\theta$ (e.g., a Wald test); and
5. a threshold $\alpha$ below which statistical significance is achieved.

$r$ is the parameter of inferential interest: it is the fraction of the initial population remaining at the end of the study period. The null hypothesis of interest is that of no change over the study period $\mathbb{H}_0 : r = 1$, which is equivalent to a nill null hypothesis (on a log scale): $\mathbb{H}_0 : \log r = 0$. To perform Monte Carlo simulations, a data-generating mechanism wherein the parameter $r$ intervenes, must be specified. We made the following assumptions.

1. Monitoring relies on a temporal sampling scheme having a total of $T$ ($T \geq 3$) sampling occasions evenly spaced at times $t \in [1{:}T]$;
2. each sampling occasion yields an abundance/density estimate $\hat{y}_t$ with an upper bound for the magnitude of their CV $cv_{y_t}$;
3. the response variable is the ratio $\hat{p}_t = \dfrac{\hat{y}_t}{\hat{y}_1}$ for all $t \in [1{:}T]$;

4. the observed values $\hat{p}_t$ follow a log-normal distribution; and

5. the true values are $p_t = r^{\frac{t-1}{T-1}}$.

With the above specification of the data-generating mechanism, it can be checked that (for $r > 0$):

$$
\begin{cases}
t = 1 & p_1 = r^{\frac{1-1}{T-1}} = r^0 = 1 \\
t = T & p_T = r^{\frac{T-1}{T-1}} = r^1 = r
\end{cases}
$$

We, thus, assumed that data $\hat{y}_t$ are collected on each sampling occasion $t$: these data may be (relative) biomass, abundance, or density. The ratio of each datum to the first datum is then computed, and the dimensionless fractions $\hat{p}_t$ resulting from these simple computations will be used to infer a trend.

## Inference strategy

The true values of $p_t$, the proportions of the population at time $t$ relative to the baseline at $t_1$, are given by the following model:

$$
p_t = r^{\frac{t-1}{T-1}} \tag{1}
$$

The parameter $r$ is represents the fraction of the initial population remaining at the end of the study period $T$. For example, $r = \frac{1}{2}$ means the halving of the initial population, or a 50% decrease over the study period. Taking the logarithm transform of Eq (1) yields:

$$
\log p_t = \log r^{\frac{t-1}{T-1}} = \frac{t-1}{T-1} \times \log r = x_t \times \beta \tag{2}
$$

where

$$
\begin{cases}
x_t = \dfrac{t-1}{T-1} \\[2mm]
\beta = \log r
\end{cases} \tag{3}
$$

Equations (2) and (3) suggest regressing the logarithm of the observed proportions $\log \hat{p}_t$ against $x_t$ to estimate $r = e^\beta$. This amounts to a linear regression with no intercept (starting from 0 at $t = 1$) and a linear trend $\beta$ over the study period. The parameter of inferential interest $r$ is related to this trend sensu *Link & Sauer (1997)*: "the percentage change (in abundance) over a specified time period". This choice of removing the intercept by modeling the ratios of abundance relative to the first estimate is highly unconventional as noted by a reviewer. Our focus on short time-series with limited information in the data to estimate many parameters motivates a desire to limit that number of parameters as much as possible. This choice is expected to increase statistical power simply by virtue of having one less parameter to estimate. Anchoring the regression line to the origin (zero) conforms to some European conservation instruments such as the HD and MSFD where conservation goals are framed with respect to a baseline, understood as "the starting

point (a certain date or state) against which the changes in the condition of a variable or a set of variables are measured" (*European Environmental Agency, 2015*).

## Simulation scenarios

We did not assume any relationship between true abundance and CV as in *Gerrodette (1987, 1991)* or *Taylor et al. (2007)*. CVs may be under the control of researchers during the planning of a survey targeting a single species. However, some surveys may collect data on several species groups to augment cost-effectiveness (*Lambert et al., 2019*): in this setting it becomes more difficult to jointly achieve a desired precision across a broad panel of species with for example, different behavior. In this setting, which is encouraged for cost-effective sampling of the marine environment, although a focal species may be a particular interest, data on other species will also be collected and the associated CVs of their estimated abundances may be viewed as random variables. Accordingly to this view, we generated CVs for abundance estimates $\hat{y}_t$ randomly from a uniform distribution. Coefficients of variation smaller than 0.1 are not common in the literature on marine mammals (*Taylor et al., 2007*), and we considered this lower bound to be the best precision to be realistically attainable with line transect surveys. CVs for marine mammal abundances can be large (*Taylor et al., 2007*). To assess the impact of the precision of estimates on detecting a trend, we varied the upper bound between 0.1 and 0.5 by 0.1 increment when simulating data. Thus 5 scenarios relating to data quality (abundance/density estimates with CVs of exactly 0.1, between 0.1 and 0.2; between 0.1 and 0.3; between 0.1 and 0.4; and between 0.1 and 0.5) were investigated.

We varied the value of $r$ ($= e^{\beta}$), the parameter of inferential interest, between 0.5 (halving of the population over the study period $T$) and 0.99 (a 1% population decrease over the study period $T$). We did not consider declines larger than 50% as these are more readily detected (*Taylor et al., 2007*), and thus focused on ambiguous cases. Finally, the length of the study period varied between 3 and 30 by increment of 1. The chosen range of values for $r$ is nevertheless broad and aligned with current goals: in simulating data, we have control over the length of the study period $T$ and $r$, the overall decline expressed as a fraction of the initial abundance. From these two parameters, the annual rate of change can be derived. European management targets are often framed with respect to the annual rate of decline: within the HD, *Bijlsma et al. (2019)* suggested a large decline to be equivalent to a loss of more than 1% per year (page 16). This corresponds to an overall decline of $\approx$ 5, 10, 18 and 26% over 5, 10, 20 and 30 years. These values are well within the range considered in our simulations.

There were $5 \times 28 \times 38 = 5,320$ different scenarios. For each of these, 10,000 data sets were simulated (see Supplemental Materials for full details and $R$ code).

## Estimation: unregularized (a.k.a. classical) and regularized

We log-transformed the simulated data before analysis with linear models. The time variable was scaled to range between 0 (start of the study period) to 1 (end of the study period). We consider a simple regression with no intercept as the first datum $\hat{p}_1$ equals 1 by design, and log(1) = 0. There was one slope parameter, $\beta$, to estimate (from at least 3 data

points). Although CVs $cv_{\hat{y}_t}$ were used to simulate data, we did not use this piece of information in the analysis to reflect a situation in which estimates may be available with only a vague idea of their precision. We thus assumed pragmatically that some abundance estimates can be available but not necessarily with their associated uncertainties in a quantitative form.

We used the default function `glm` from R (*R Core Team, 2018*). This function returns Maximum Likelihood estimates of β. These estimates $\hat{\beta}^{ML}$ are by default unregularized and may be improved with prior information, especially in data sparse settings (*Gelman et al., 2014*). Priors need not reflect any 'subjectivity' but rather characterizes transparent decisions (*Gelman & Hennig, 2017*) with respect to the many possible analytical choices that may be available (so-called "researchers degrees of freedom", *Simmons, Nelson & Simonsohn (2011)*; and "garden of forking paths" *Gelman & Loken (2013)*; see *Fraser et al. (2018)* for ecological research). We adhere to the view of *Gelman & Hennig (2017)* and see the prior in the context of trend estimation as a technical device (page 991) to "exploit a sensitivity-stability trade-off: they (the priors) stabilize estimates and predictions by making fitted models less sensitive to certain details of the data" (*Gelman & Shalizi, 2013*). In addition to `glm`, we used the `bayesglm` function implemented in R package `arm` (*Gelman & Su, 2018*) to obtain regularized estimates $\hat{\beta}^{reg}$ (see the Supplemental Materials for the associated R code). The prior helps to stabilize estimates and robustifies results against sampling noise in the data: in this sense one obtains regularized estimates. Our Monte Carlo study is an instance of "calibrated Bayes" sensu *Little (2006)* in which we are evaluating the frequentist properties of Bayesian methods. We considered two priors: an informative prior and a weakly-informative one (Fig. 1).

The informative prior was chosen to cover a priori a range associated with the halving or doubling of the population over the study period: we chose a symmetric normal prior (on a logarithmic scale) centered on 0, and set the scale parameter to log(2)/2 (Figs. 1A and 1B). The weakly-informative prior was a Cauchy distribution proposed by *Cook, Fúquene & Pericchi (2011)* which translated the idea that the null (no decline) is assumed a priori true with odds of 39:1. Its location parameter was accordingly set to 0 and its scale parameter was set to $-\frac{\log(2)}{\tan(\pi(\xi - \frac{1}{2}))}$ where ξ is a small (skeptical) probability that $r$ (β) is different from 1 (0; *Cook, Fúquene & Pericchi, 2011*). The weakly-informative prior with ξ = 0.025 is shown in Figs. 1C and 1D. These two priors will regularize estimates of the unknown trend parameter $r$ (β) with shrinkage toward the value 1 (0), thereby acting as regularization devices against idiosyncratic noise in the data. However, in the case of the weakly-informative prior, if the signal in the data is strong, the prior will give way and exerts little shrinkage toward the null. Thus, priors as regularization devices will yield biased estimates, with the direction of the bias known (bias toward the prior location parameter) but with an increased precision. In contrast, unregularized estimates may be very biased and imprecise.

For each of the 10,000 simulated data sets, the p value associated with $\mathcal{H}_0$ was stored (see the Supplemental Materials for the associated R code). Statistical power is the frequency with which a false $\mathcal{H}_0$ is rejected at the chosen significance level. We considered

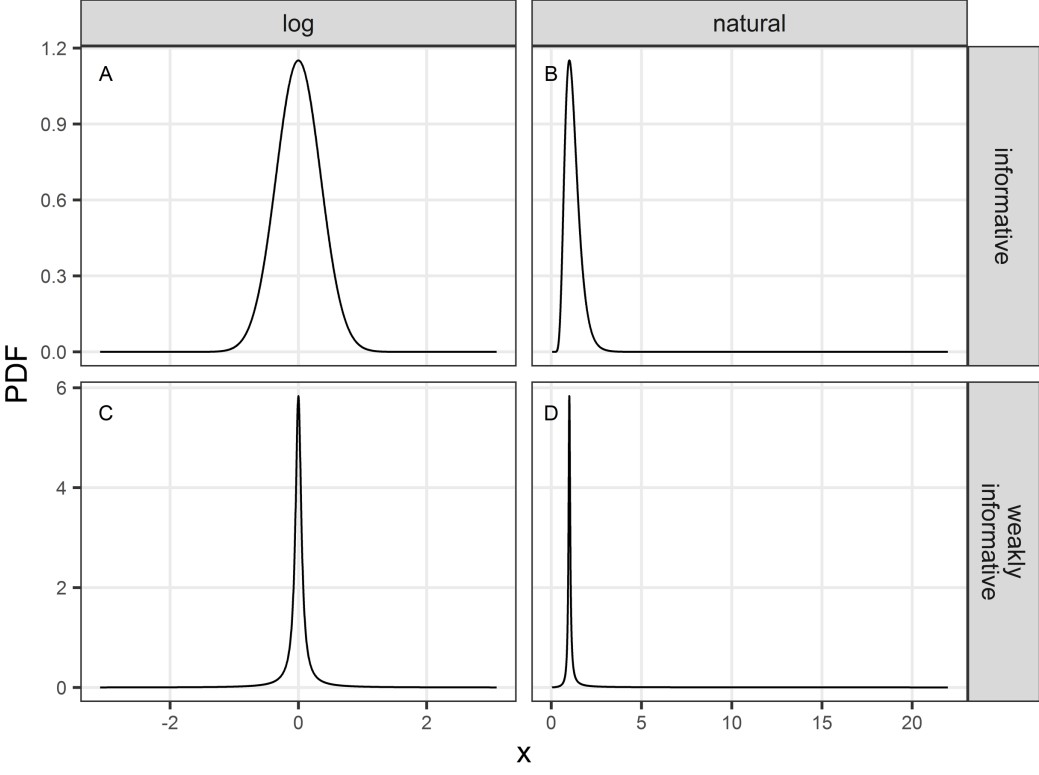

**Figure 1 Probability density function (PDF) of the informative (A and B) and weakly-informative (C and D) priors used in regularized regression approaches.** PDF are shown either on a logarithmic (A and C) or natural scale (B and D).      

two significance levels: 0.05% and 0.20%. The relaxed significance level of 0.20% was suggested by *ICES (2008*, *2010)* to increase statistical power and to have equal probability of Type I and II errors, and in line with a precautionary principle. However, this recommendation equates level of statistical significance with type-I error rates: it confuses statistical significance *à la* Fisher with type-I error rate *à la* Neyman–Pearson (*Hubbard & Bayarri, 2003*; *Christensen, 2005*). Nonetheless, the hybrid approach, even if it confuses significance and type-I error rate, is widely used in conservation decisions and needs to be assessed in this context. We estimated the type-I error rates of our proposed approach by running Monte Carlo simulations with $\beta = 0$; that is when the null hypothesis of no decline over the study period is true. With our comprehensive factorial design crossing (a) sample size (study length), (b) effect size (decline magnitude), (c) data precision (CV) and (d) statistical approach (regularized regression or not), we thus assessed power, statistical significance and computed associated confidence intervals for $\hat{r}$. We assessed confidence interval coverage, both unconditional and conditional on statistical significance. Finally, we assessed the type-S and type-M error rates of statistically significant estimates.

## Case studies

We applied our proposed regularized approach on a handful of recent case studies in European waters. We collected 132 abundance or density estimates from published

**Table 1 Case studies, and associated references, for regularized estimation of population trends of cetacean species in European waters.** The column "Design" refers to the design of the data collection scheme: Distance Sampling (DS) or Capture-Mark-Recapture (CMR). The vaquita is included for illustrative purposes (see also Supplemental Materials).

| Species | Scientific name | Period | Area | Season | Sample size | Design | References |
|---|---|---|---|---|---|---|---|
| Fin whale | *Balaneoptera physalus* | 2007–2016 | Wider Bay of Biscay | Autumn | 6 | DS | *García-Barón et al. (2019)* |
| Minke whale | *Balaneoptera acutorostrata* | 2004–2016 | Bay of Biscay | Spring | 13 | DS | *Authier et al. (2018)* |
| | | 1989–2016 | North Sea | Summer | 10 | DS | *ICES (2017)* |
| Risso's dolphin | *Grampus griseus* | 2004–2016 | Bay of Biscay | Spring | 13 | DS | *Authier et al. (2018)* |
| Long-finned pilot whale | *Globicephala melas* | 2004–2016 | Bay of Biscay | Spring | 13 | DS | *Authier et al. (2018)* |
| Bottlenose dolphin | *Tursiops truncatus* | 2004–2016 | Bay of Biscay | Spring | 13 | DS | *Authier et al. (2018)* |
| | | 2005–2016 | Wider bay of Cardigan | Year-round | 12 | CMR | *Lohrengel et al. (2018)* |
| | | 2010–2017 | Gulf of Saint Malo | Year-round | 8 | CMR | *Grimaud, Galy & Couet (2019)* |
| Common dolphin | *Delphinus delphis* | 2004–2016 | Bay of Biscay | Spring | 13 | DS | *Authier et al. (2018)* |
| | | 2007–2016 | Iberian Coasts | Spring | 10 | DS | *Saavedra et al. (2018)* |
| Striped dolphin | *Stenella coeruleoalba* | 2004–2016 | Bay of Biscay | Spring | 13 | DS | *Authier et al. (2018)* |
| White-beaked dolphin | *Lagenorhynchus albirostris* | 1994–2016 | North Sea | Summer | 3 | DS | *Hammond et al. (2017)* |
| Harbour porpoise | *Phocoena phocoena* | 1994–2016 | North Sea | Summer | 3 | DS | *Hammond et al. (2017)* |
| Vaquita | *Phocoena sinus* | 1997–2016 | Sea of Cortez | Summer | 4 | DS | *Taylor et al. (2017)* |

research or reports (Table 1). The number of available estimates for trend estimation varied between 3 and 13. We included data on the vaquita for illustration of an unambiguous case of a dramatic decline (see also *Gerrodette, 2011*, see Supplemental Materials). For each case study, we estimated trends with both an unregularized and regularized approach. We investigated the stability of estimates of annual growth rate with increasing sample size to mimic an ongoing study in which data are collected at regular intervals. Annual growth rates $\hat{r}_a$ were computed from trend estimates $\hat{r}$ scaled back to an annual timeframe: $\hat{r}_a = \sqrt[T]{\hat{r}}$.

## RESULTS

### Type-I error

Empirical Type-I error rates were not equal to the chosen significance levels (Fig. 2). For all sample sizes, the unregularized approach (classical generalized linear model) had a Type-I error rate of at least 10% when the significance level was set to 5%; and a Type-I error rate of at least 30% when the significance level was set to 20%. The regularized approach with an informative prior had a Type-I error rate close to the chosen significance level for small sample size only (≤10), and inflated Type-I error rate with increasing sample size. The regularized approach with a weakly-informative prior had a Type-I error rate less than 10% when significance was set at 5%, and close to 5% for CVs less

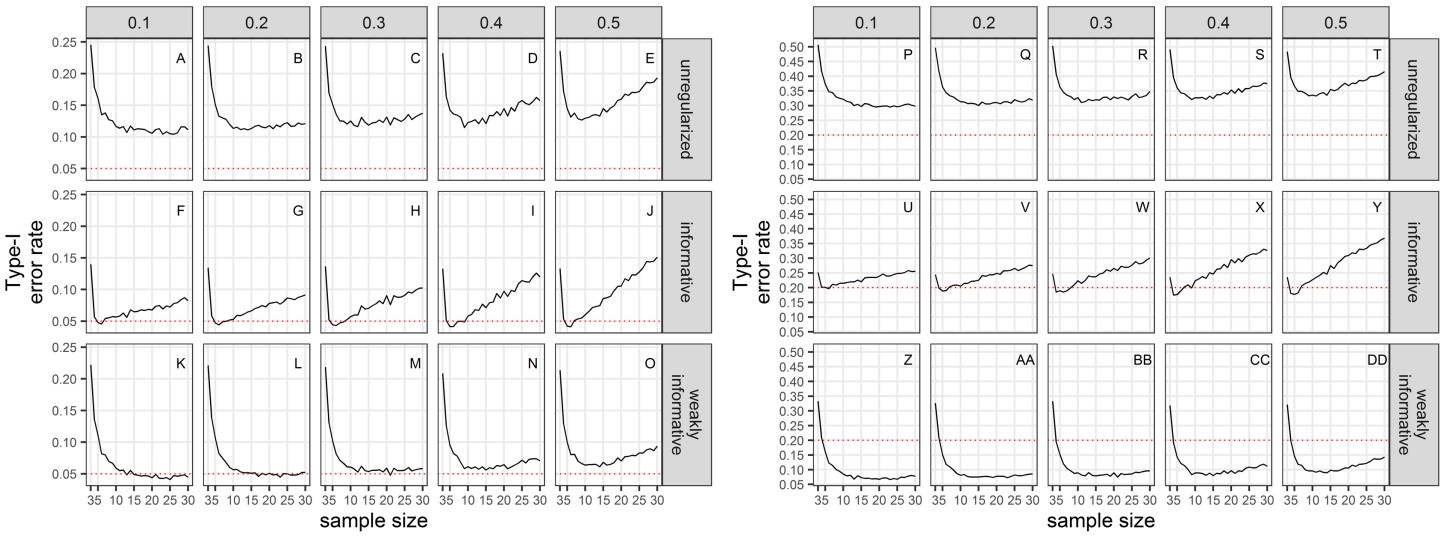

**Figure 2 Type-I error rate of a two-tailed test of no trend over a study period T, with a significance level set to 5% (A–O) or 20% (P–DD).** The dotted red line materializes the chosen significance level.

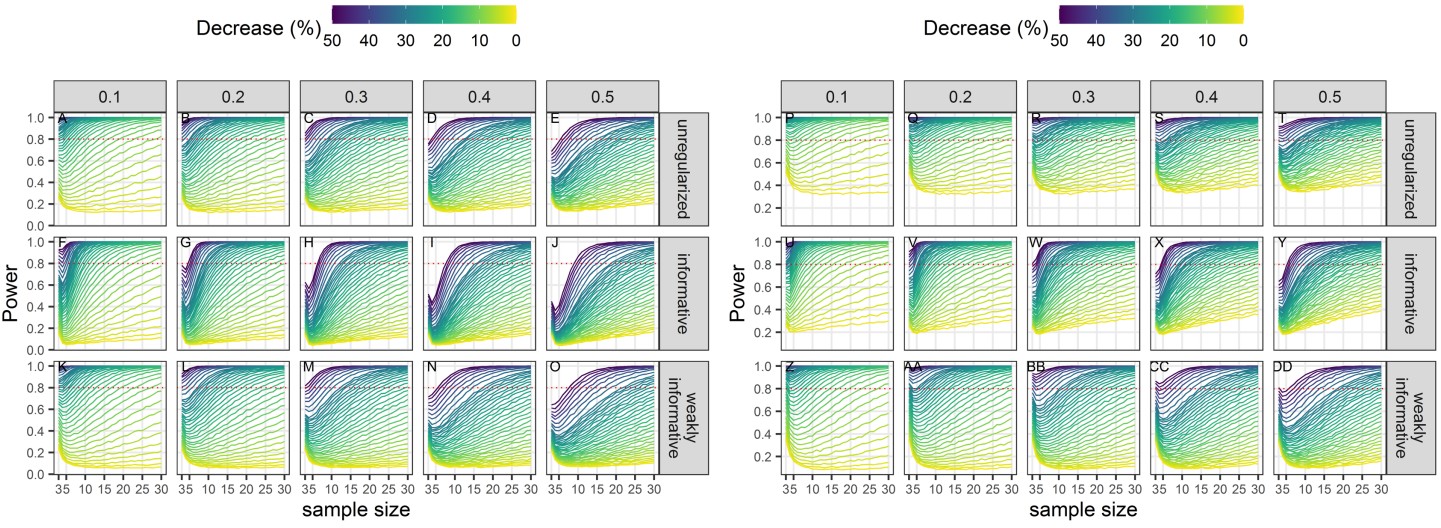

**Figure 3 Power of a two-tailed test with a significance level set to 5% (A–O) or 20% (P–DD) to detect a population decline over a study period *T*.** Each column corresponds to a different assumption with respect to the precision of abundance estimates on which the trend is inferred.

than 0.4. When significance was set to 20%, type-I error rates were below 20% for sample size ≥ 4.

## Power

Power to detect a statistically significant trend increased with sample size, magnitude of decline and precision of data (Fig. 3). Regularized approaches were less powerful than the unregularized one, with the greatest loss of power associated with using an informative prior on a short time-series of noisy estimates. For all approaches, the power to detect

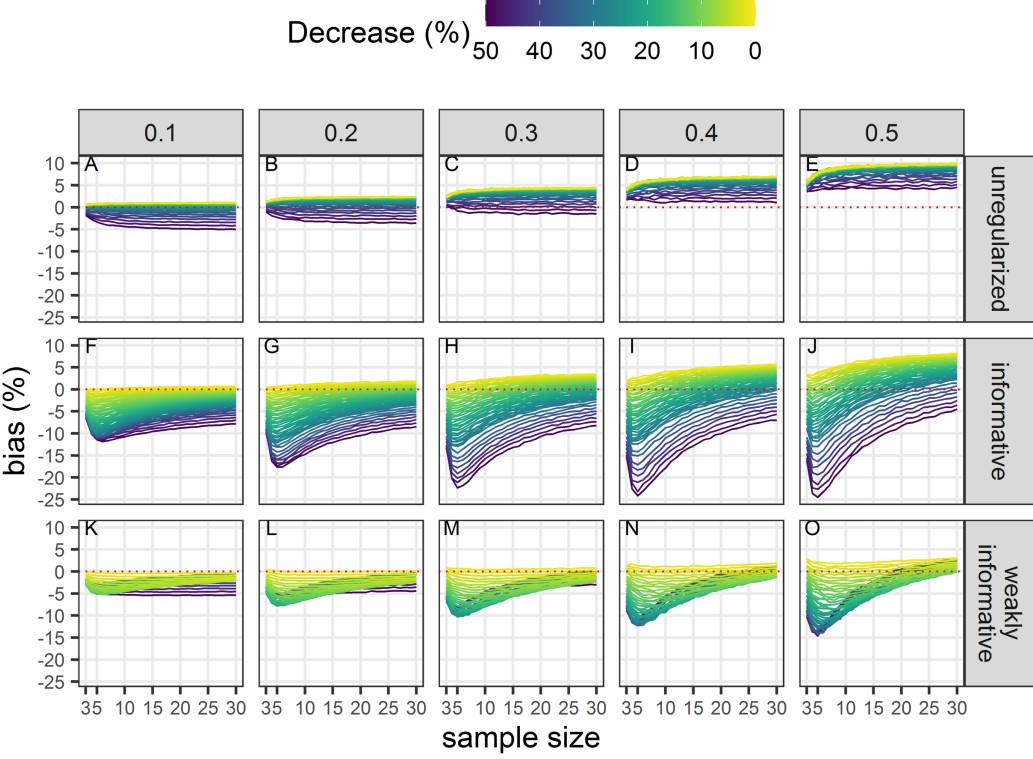

**Figure 4  Bias in the estimated population decline.** Each column corresponds to a different assumption with respect to the precision of abundance estimates on which the trend is inferred. The dotted red line materializes no bias. All estimates, statistically significant or not, are included in this assessment. (A–E) Results from unregularized regression; (F–J) results from regression with an informative prior; and (K–O) results from regression with a weakly-informative prior.

decline of less than 5% over the study period was low (less than 0.5) to very low (less than 0.2).

## Bias unconditional on statistical significance

All estimates were biased (Fig. 4) with the magnitude of the bias depending on sample size, magnitude of decline and precision of data. With unregularized regression, bias was mostly a function of data precision with an increasing positive bias (that is an overestimation of decline) with an increasing CV. The range of bias was largest with regularized regression with an informative prior. This approach yielded underestimates of trends when sample size was small, true decline was small and precision was low. It resulted in overestimates with large sample size and low data precision. Regularized regression with a weakly-informative prior resulted in estimates with the lowest bias, with a bias that was mainly negative (that is, underestimates) except for large sample size and imprecise data.

## Coverage unconditional on statistical significance

Empirical coverage of 95% confidence intervals was never at its nominal level (Fig. 5). Coverage improved with smaller CVs but was especially low with regularized regression

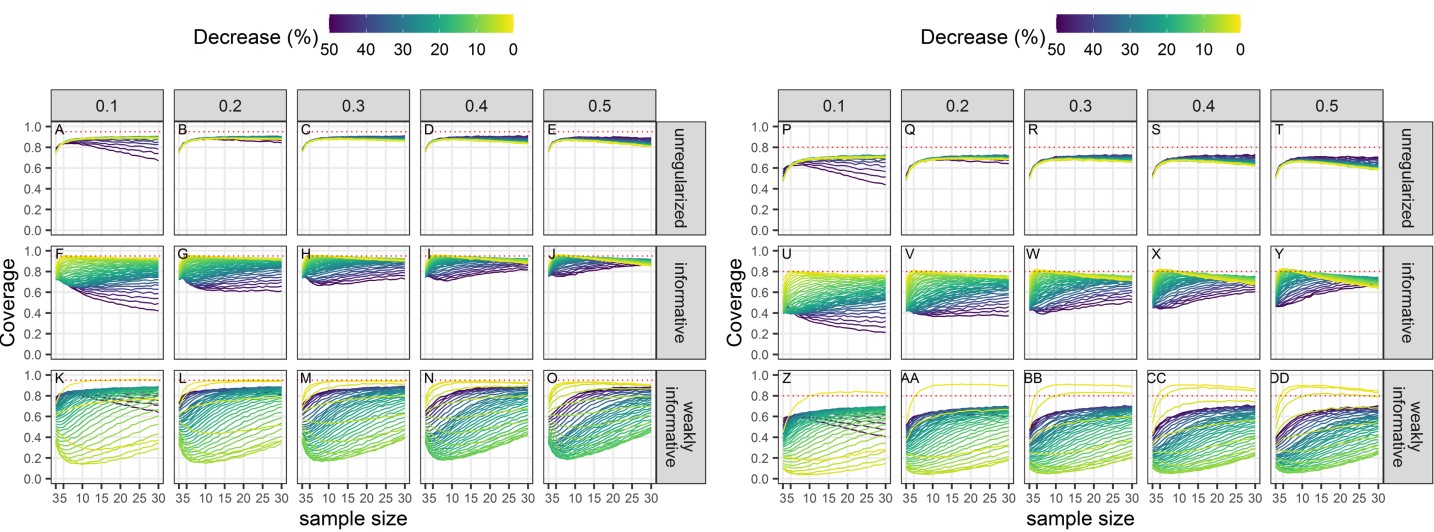

**Figure 5 Empirical coverage of confidence intervals for the estimated population decline using a significance level of 5% (A–O) or 20% (P–DD).** Each column corresponds to a different assumption with respect to the precision of abundance estimates on which the trend is estimated. (A–E) and (P–T) Results from unregularized regression; (F–J) and (U–Y) results from regression with an informative prior; (K–O) and (Z–DD) results from regression with a weakly-informative prior.

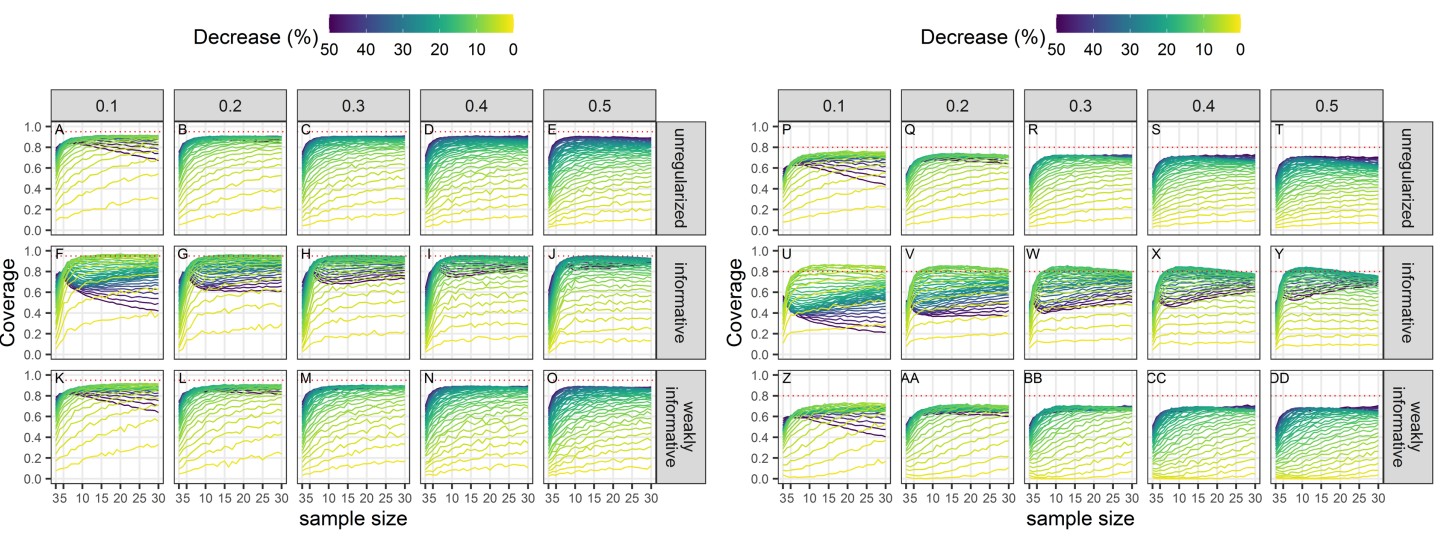

**Figure 6 Empirical coverage of confidence intervals for the estimated statistically significant population decline using a significance level of 5% (A–O) or 20% (P–DD).** Each column corresponds to a different assumption with respect to the precision of abundance estimates on which the trend is estimated. (A–E) and (P–T) Results from unregularized regression; (F–J) and (U–Y) results from regression with an informative prior; (K–O) and (Z–DD) results from regression with a weakly-informative prior.

with a weakly-informative prior except when negligible trends (of the order of 1 or 2% over the study period) were being estimated.

## Coverage conditional on statistical significance

Empirical coverage of 95% confidence intervals of statistical significant estimates was not, in general, at its nominal level (Fig. 6). Coverage was closest to its nominal value for

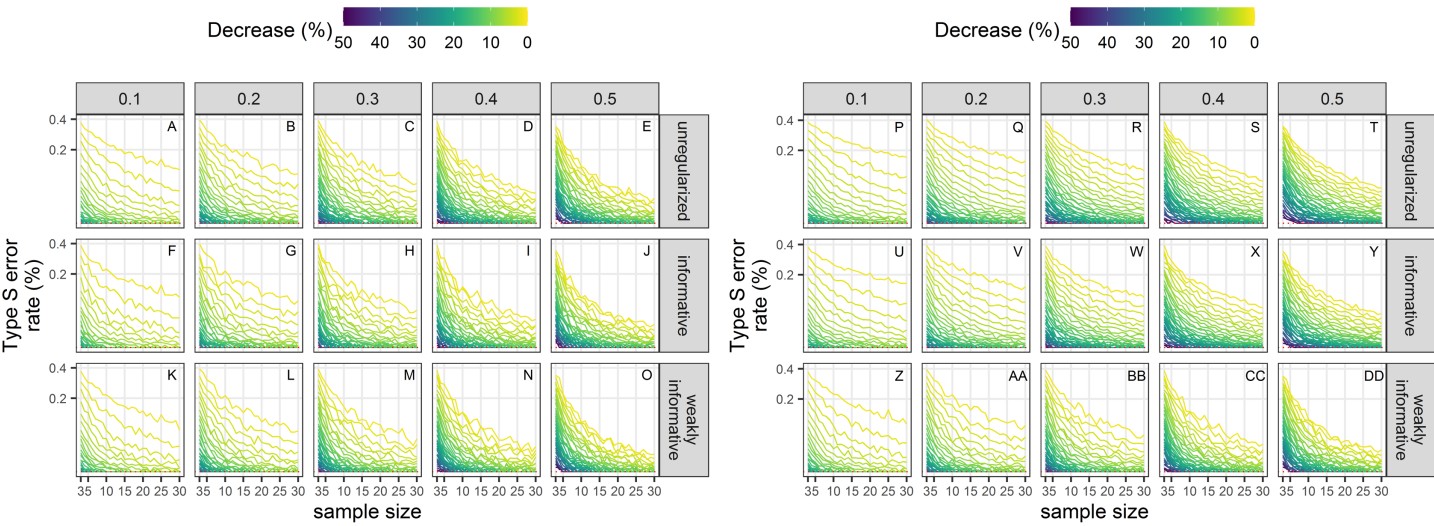

**Figure 7 Empirical Type-S error rates associated with a significance level of 5% (A–O) or 20% (P–DD).** Each column corresponds to a different assumption with respect to the precision of abundance estimates on which the trend is estimated. Note the square-root scale on the y-axis. (A–E) and (P–T) Results from unregularized regression; (F–J) and (U–Y) results from regression with an informative prior; (K–O) and (Z–DD) results from regression with a weakly-informative prior.

regularized regression with an informative prior. There was little difference between an unregularized approach and a regularized one with a weakly-informative prior.

## Type-S error rates

Type-S error rates were larger with small sample size and small magnitude of decline (Fig. 7). When trying to detect a small decline with precise data (CV = 0.1), type-S error rates were the largest suggesting that a small amount of noise in estimates could easily lead to spurious inference of an increase in this setting, unless sample size was large. Regularized approaches had lower type-S error rates than an unregularized one. Setting the significance level to 20% instead of the classical 5% resulted, *ceteris paribus*, in a small increased probability of type-S error.

## Type-M error rates

Exaggeration factors (Type-M error rates) were the largest for regularized regression with an informative prior, but similar between unregularized regression and regularized regression with an weakly-informative prior (Fig. 8). The latter two approaches tended to underestimate a statistically significant trend, especially with imprecise data.

The results are summarized in Table 2. The main differences between the different approaches are with respect to bias, type-I and type-M error rates. The regularized approach with a weakly-informative prior had a consistent underestimation bias and a type-I error rate under control. In contrast, the unregularized approach and the regularized approach with an informative prior could yield over- or under-estimates, and had an inflated type-I error rate.

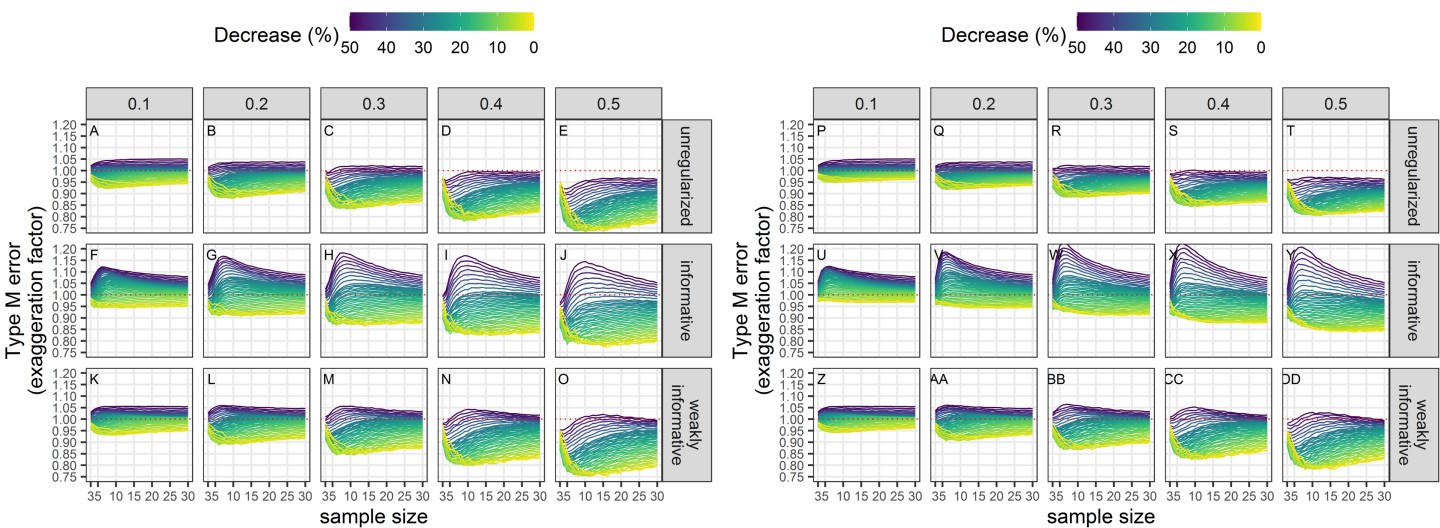

**Figure 8 Empirical exaggeration factors (a.k.a. Type-M error rates) associated with a significance level of 5% (A–O) or 20% (P–DD).** Each column corresponds to a different assumption with respect to the precision of abundance estimates on which the trend is estimated. (A–E) and (P–T) Results from unregularized regression; (F–J) and (U–Y) results from regression with an informative prior; (K–O) and (Z–DD) results from regression with a weakly-informative prior.

**Table 2 Comparing results from the three approaches to estimate and detect a trend across the different scenarios.** α is the threshold for statistical significance.

|  | Unregularized | Informative | Weakly-informative |
|---|---|---|---|
| **Unconditional on statistical significance** | | | |
| Bias | over- or under-estimation | | underestimation |
| Coverage | includes the true value less than $1 - \alpha$ times | | |
| **Conditional on statistical significance** | | | |
| Power | <80% except for large decline | | |
| Coverage | includes the true value less than $1 - \alpha$ times | | |
| Type-I error | inflated ($>\alpha$) | | under control ($\leq\alpha$) |
| Type-S error | large for small declines | | |
| Type-M error | underestimation | over- or under-estimation | underestimation |

## Case studies

Estimates of annual growth rates for 14 populations of cetaceans were similar in magnitude across the different approaches, with the biggest differences for time-series of less than 5 data points (Figs. 9A–9L). Estimates from regularized regression approaches were somewhat attenuated, that is biased towards 1, compared to those from unregularized regression. Estimates from regularized regression approaches were also more precise, especially those with a weakly-informative prior. This increased precision would allow to reach a conclusion with respect to trend detection faster.

For the vaquita, the estimated annual growth rate was estimated at 0.88% (80% confidence interval 0.86–0.90), a figure similar that of *Taylor et al. (2017)* who estimated an annual growth rate of 0.87% (95% credible interval 0.82–0.91).
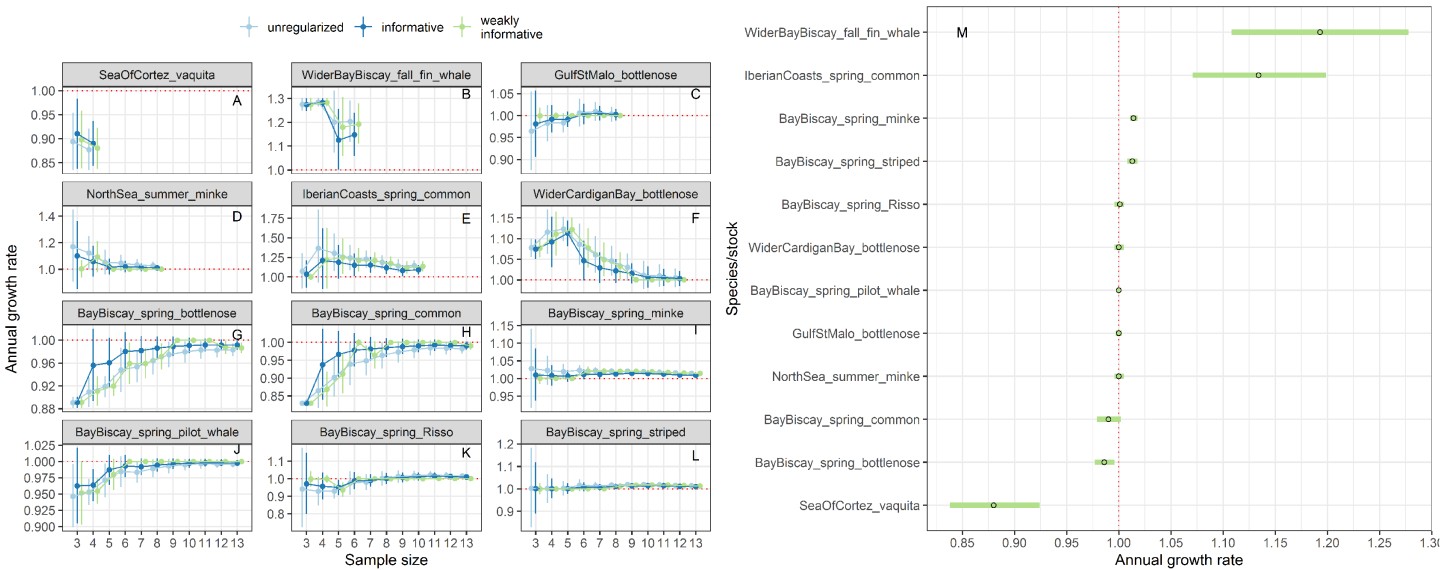

**Figure 9 Stability of trends estimates for the different approaches (A–L) and point estimates along with 80% confidence interval from a regularized regression with a weakly-informative prior (M).** Harbour porpoises and white-beaked dolphins are not depicted because only three estimates were available.

## DISCUSSION

Accurate estimation of population trend of marine mammals is a difficult and rather depressing endeavor (*Taylor et al., 2007*; *Jewell et al., 2012*). This stems from both challenging conditions at sea and the intrinsic ecology of marine mammals, including their mobility (wide areas across international borders need to be covered during sampling, with large operational costs money-wise) and their elusiveness (imperfect detection). As a result, conservationists usually have to base decisions on short time-series of noisy abundance/density estimates even when state-of-the-art methods and sampling design are used. Turning once again to the vaquita for illustrative purposes, even though its estimated abundance more than halved between 1997 and 2008, from 567 to 245; the width of the confidence intervals associated with these abundance estimates remained roughly constant at about 800 or 900 individuals (*Taylor et al., 2017*). High estimation uncertainty is endemic, except in some cases of Mark-Capture-Recapture studies.

We investigated the practical consequences of this uncertainty with respect to frequentist properties of unregularized (classical) and regularized regression approaches. The unregularized approach did not meet quality expectations: it had inflated type-I error rates compared to the customary significance level of 5%. Relaxing the latter to 20% as recommended by *ICES (2008*, *2010)* did not remedy this issue. In both cases, type-I error rates increased with sample size when uncertainty was large, a counterintuitive result which underscores that noise can easily dominate the signal in trend analyses. In contrast, regularized regression with a weakly-informative prior (*Cook, Fúquene & Pericchi, 2011*) kept type-I error under the 20% significance level in the face of large uncertainty in abundance estimates with no additional computational cost compared to the unregularized approach.

It may come as surprising from a glance at Fig. 1 that what appears as a very informative prior is actually not so, or that the default option of equating uniform with uninformative is misleading (*Dorazio, 2016*; *Gelman, Simpson & Betancourt, 2017*; *Gabry et al., 2019*). The prior is what distinguishes Bayesian from classical statistics, with the oft mentioned pros of the former approach being its ability to incorporate auxiliary information with data in an analysis (*Wade, 2000*; *Ellison, 2004*; *Clark, 2005*). This ability is not unique to the Bayesian approach (*Taper & Ponciano, 2016*) but a discussion of the so-called "statistics wars" in ecology and conservation is beyond the scope of this study (see also *Toquenaga, 2016*); suffice it to say that the prior is the price to pay for a Bayesian analysis. Few studies using Bayesian methods in ecology and evolution used informative priors in practice, but most relied on non-informative priors, meaning either uniform priors or very diffuse priors (*Dorazio, 2016*). This can have unfortunate consequences as what looks uninformative of one scale may be very informative on another (*Dorazio, 2016*; *Yamamura, 2016*). The appeal of uniform priors may stem from the desire to prevent personal idiosyncrasies of a researcher to influence analyses, that is to uphold objectivity. The adjectives "objective" and "subjective" are loaded: *Gelman & Hennig (2017)* called for avoiding using them altogether, a suggestion which triggered a lively discussion among leading statisticians (see the ≈ 70 comments published along side with *Gelman & Hennig (2017)*). For the pragmatical ecologist, the question remains: should an informative prior be used, and if so, will it convince colleagues and legislators, especially in an applied conservation context.

The choice of the null hypothesis is not benign. Implicit in choosing a null hypothesis of no effect (a nil null hypothesis) is the assumption that a type-II error (failing to detect a decline) carries less costs than a type-I error (concluding there is a trend when it is in fact nil). This scientific preference is congruent with the "innocent until proven guilty" standard in criminal law (*Noss, 1994*), but puts the burden of proof on the shoulders of conservationists. However, only dramatic declines are readily detected (*Taylor et al., 2007*) and irremediable damage or loss may occur because measures are delayed in the light of statistically insignificant declines. This shortcoming of nil null hypotheses is well known (*Noss, 1994*, *Buhl-Mortensen, 1996*), but current conservation instruments in Europe such as the Habitats Directive or the Marine Strategy Framework Directive have not taken stock of it. Here, we have carried out extensive simulations to show that type-I errors are not even minimized with standard (unregularized) regression techniques applied on realistic data for cetaceans. Only by incorporating auxiliary information in the form of a weakly-informative prior could we achieve type-I error rates congruent with the recommendations of *ICES (2008, 2010)* to equalize the probabilities of type I and type II errors. Relaxing the threshold for significance from 5% to 20% resulted in an increase of statistical power as expected, but our regularized regressions, all else being equal, were less powerful that unregularized ones (Fig. 3). However, statistical power of regularized regression with a weakly-informative prior and significance level set to 20% was similar to the power of unregularized regression with significance level set to 5% (Fig. 3). These results strongly suggests that incorporating prior information in the detection of a trend is actually better aligned with default expectations of both scientists

and legislators. It is our opinion that the pragmatical ecologist ought, in fact, to use priors and can rebutt claims of "obfuscating the challenges of data analysis" (*Ionides et al., 2017*) with an evaluation of Bayesian procedures from their long-run frequency properties (*Rubin, 1984*) similar to the evaluation we have carried out in this study.

Although type-I and type-II errors are often discussed in applied conservation studies, there are also other kinds of errors that are no less detrimental, and that go beyond the binary detection of a trend to the reliability of the estimate with respect to its sign and magnitude. These are the type-S and type-M errors of *Gelman & Tuerlinckx (2000)*, which were also the focus of our Monte Carlo study. Type-S error rates increased with decreased precision of abundance estimates in all cases, and the largest rates were associated with the smallest declines, that is, the signal that was the hardest to detect. Type-S error rates of regularized regression approaches were smaller than those of an unregularized one: using a prior was beneficial with respect to the accuracy of the inference. With a prior, the chance of reaching a wrong conclusion, that is, inferring an increase when in fact there was a decrease, was lowered, but could be as high as 20% with short and very noisy time-series of abundance estimates. Results with respect to type-M error rates were more contrasted: a decrease in precision resulted in statistically significant estimates being underestimates of the true magnitude of the decline for unregularized and weakly-informative regularized regression. This underestimation was surprising as we were expecting in fact an exaggeration of effect sizes due to conditioning on statistical significance (*Lane & Dunlap, 1978*; *Gillett, 1996*; *Gelman & Carlin, 2014*). This may be due to the fact that we removed the intercept in our analyses, thus anchoring trend estimates to the first value in the time series. Consequently, this first value can have a large influence on inferences. This exaggeration of effect sizes was, however, apparent with regularized regression with an informative prior: estimates were too large when the true decline was also large (see Supplemental Information). The reason for this was statistical significance: with regularized regression with an informative prior, the statistically significant estimates were more biased away from 0 on average compared to the other approaches. Taken together, empirical type-S and type-M error rates suggest that the best trade-off is reached when a weakly-informative prior is used.

Confidence intervals associated with a given level, say 95%, are supposed to contain the true value of an unknown parameter with the same long-term frequency. We investigated whether the empirical coverage of confidence intervals around a trend were at their nominal value, and found that it almost never was. That coverage of confidence interval may differ from the nominal value may, again, come as surprising to ecologists (*Agresti & Coull, 1998*). In our study, coverage was always lower with unregularized regression. The effect of conditioning on statistical significance was pronounced: when trying to estimate small declines (in magnitude), coverage was close to the nominal level with regularized approaches if one did not condition on statistical significance, whereas coverage dropped dramatically when only statistically significant estimates were retained. Overall, conditioning on statistical significance gave similar results between an unregularized approach and a regularized one with a weakly-informative prior, although coverage was smaller for the latter than for the former, all else being equal.

This reduced coverage is expected since the weakly-informative prior is introducing bias by definition: it shrinks a priori the trend toward a nil value. Moreover, the weakly-informative prior has a small dispersion, thereby encouraging narrow confidence intervals but the choice of a heavy-tailed distribution such as the Cauchy means that in the case of a data-prior conflict, the data will dominate the analysis. Overcoming the prior when there is a conflict with data entails a loss of power (Fig. 3), but this loss was modest and more than offset by other *desiderata*. For instance, confidence intervals derived for a regularized approach with a weakly-informative prior were narrower, to the effects that they could exclude the true value more often (reduced coverage), even though the overall bias and type-M error rates of this approach were no worse than alternative estimators. With respect to bias, it was always negative or close to zero with the regularized approach with a weakly-informative prior, whereas the sign of the bias could be either positive or negative with the other two approaches.

We believe that our Monte Carlo study clearly points to the superiority of incorporating weak information in the form of a prior to carry out the difficult task of detecting and estimating a decline in short and noisy time-series of abundance or density estimates. A question remains vis-à-vis real case studies wherein the data-generating mechanism is unknown, which we tackled by looking at a handful of recent studies of cetaceans in European waters (Table 1). The differences between the unregularized and weakly-informative regularized approaches were small (Fig. 9), with the latter producing estimates with narrower confidence intervals, as expected. Similar conclusions would be reached for all case studies we considered irrespective of methodological choice: all methods converged to the similar estimate values and associated standard errors with increasing sample size. Thus, in practice, the same conclusions would have been reached, but the weakly-informative regularized approach offers more empirical guarantees with respect to its long-term performance, especially with short time-series. The vaquita again provides a point of reference for population growth rate: a vaquitan decline is one of more than 10% per year, meaning than the species with such a decline will be on the same path as the vaquita (an example of a vaquitan decline in the terrestrial realm is that of Grauer's Gorilla, *Gorilla beringei graueri*; *Plumptre et al., 2016*). The high annual growth rate for fin whales in the wider Bay of Biscay, and for common dolphins of the Iberian coasts suggested immigration and open populations (*Saavedra et al., 2018*; *García-Barón et al., 2019*). Compared to the original published results, inferences were similar except for those on population trends of common and bottlenose dolphins in the Bay of Biscay during spring. *Authier et al. (2018)* did not find a decrease for these two species, but their analysis was different and did not estimate a trend as it relied on a Dynamic Factor Analysis to infer a common trajectory in relative abundance for a panel of 23 species of marine megafauna. Here, our analysis focused on detecting and estimating a trend on a species per species basis, and the discrepancy is due to the first estimate in the time-series for both common and bottlenose dolphins being the largest. This testifies to the high leverage that the first datum can have, and illustrates further that the choice of the baseline is critical (*ICES, 2010*). Information on "edenic" baselines, referring to abundance levels before any anthropogenic alterations, are difficult to document, or

entirely lacking (*Lotze & Worm, 2009*; *McClenachan, Ferretti & Baum, 2012*). *Kopf et al. (2015)* suggested the use of "anthropocene baselines" as a "dynamic point of reference for human-dominated ecosystems" rather than focusing on a fixed point of reference from pristine (that is, pre-industrial) conditions which are largely unknown. This concept of "anthropocene baselines" aligns well with requirements of the Marine Strategy Framework Directive, the latest normative conservation instrument for marine ecosystems in Europe.

Norms are "standards of appropriate behavior for actors (… and) reflect temporally, socially, and materially generalized behavior expectations in a given social group" (*Deitelhoff & Zimmermann, 2020*). Their ultimate purpose is to solve problems of collective action, and statistical significance does qualify as a norm for reaching a decision in the face of uncertainty in applied conservation. This paper hence deals with contesting a norm, not in challenging its validity but its current application. Validity concerns about reliance on statistical significance have been detailed elsewhere: see for example *Gerrodette (2011)* with respect to applied conservation, *Amrhein, Greenland & McShane (2019)* for science in general; and for statistical science *Wasserstein & Lazar (2016)*, *Wasserstein, Schirm & Lazar (2019)* (along other contributions on the topic in the 73th issue of *The American Statistician*). Our concern here stems from applicatory conditions of the current norms in conservation instruments. For example, *ICES (2010)*'s suggestion to relax the threshold for statistical significance to 20% enacts a challenge to the (usually) unquestioned default figure of 5%, a default which reflects its internalization by stakeholders, and hence its validity (*Deitelhoff & Zimmermann, 2020*). We heeded that call and further challenged the current norm by considering Bayesian regularization to improve the detection and estimation of trends. To demonstrate the adequacy and relevance for conservation of such a norm change we carried out a comprehensive study by means of Monte Carlo simulations to assess the long-run frequency properties of both the unregularized (statu quo) and regularized (challenger) statistical procedures used for trend estimation in the spirit of Calibrated Bayes (*Little, 2006*, *2011*). We are thus not advocating a norm decay with wolf-in-sheep's-clothing priors, but a very precise norm change consistent with *ICES (2010)*'s suggestion. We have shown that setting the threshold for statistical significance to 20% and using regularized regression with a specific weakly-informative prior provide a superior alternative to current practices according to the same criteria (type-I, type-II error rates, bias, coverage), along with additional ones (type-M, type-S error rates), that are routinely invoked to justify the current norm. Furthermore, regularization with a weakly-informative prior was able to yield estimates with similar or less bias than the unregularized approach, even when estimates were not statistically significant (Fig. 4). Moreover, the bias was always negative thus giving conservative estimates of declines and avoiding an exaggeration of the magnitude of declines. Although this slight underestimation of a decline may seem to contravene a precautionary approach, it can nevertheless be taken into account in conservation decisions because its direction is known and systematic, which means in practice that uncertainty is reduced (e.g., "the decline was at least of $x$%"). In order to truly abide

by a precautionary approach, regularized estimates should be corrected for this underestimation bias, and our simulations provide correction factors to re-calibrate empirical trend estimates irrespective of statistical significance (Fig. 4).

## CONCLUSIONS

Applied conservation faces many challenges in the Anthropocene, ranging from climate change to the dire impact of ever-expanding anthropogenic activities (*Sutherland et al., 2019*). In the face of high uncertainty and (more often than not) few data, ecologists must base decisions on trends detected and estimated from short and noisy time-series, where the usual (asymptotics) guarantees no longer hold. Focusing on simple methods available from freely available software, we investigated weakly-informative regularized regression as a tool to disentangle a meaningful signal, a population trend, from measurement error. Our philosophical outlook was instrumentalist in that we have no doubt that we are proposing a very simple model but were interested in the quality of inferences drawn from this undoubtedly mis-specified model given the data available right now to ecologists and conservationists. In particular, we ignored some information in the analyses (e.g., the exact precision of abundance estimates) but not entirely as we considered several possible and realistic ranges. We used the vaquita, a small endemic cetacean on the brink of extinction, mostly for illustrative purposes (*Gerrodette, 2011*), but also looked at other species in the context of the European conservation instruments.

It is worth keeping in mind that, in general, a trend is a crude simplification and is a convenient summary statistic to understand and to communicate: it provides a counterfactual of what would have been the annual growth rate, had it been constant over the study period (e.g., Fig. 8). The longer the time-series, the less realistic this fiction becomes. With long time-series, say more than 10–20, more complex methods, in particular state-space model methods (*De Valpine & Hastings, 2003*; *Knape, Jonzén & Sköld, 2011*), are appropriate as they can take into account both process and measurement errors and make better use of all the available information. However, the target of the analysis has now shifted from estimating a population trend to estimating the whole trajectory of the population (*Link & Sauer, 1997*). It is also important to realize that (i) the null hypothesis of no trend over time is, strictly speaking, always false; and (ii) a trend analysis cannot, in general, elucidate the cause of the decline. For that latter endeavor, both experiments and modeling the whole population trajectory with state-space models is better suited as it can leverage process-level variations to identify causes (*Nichols & Williams, 2006*; *Hovestadt & Nowicki, 2008*; *Knape, Jonzén & Sköld, 2011*).

We investigated the statistical guarantees of a method, linear regression with regularization, in a very circumscribed context, that of estimating a trend relative to a baseline in short and noisy time-series. Furthermore, we took several unconventional steps (i.e., we used some researcher degrees of freedom sensu *Simmons, Nelson & Simonsohn (2011)*), including anchoring the regression line at the origin while working relative to a baseline; and ignoring information on estimate uncertainty at the analysis stage.

These choices should not be viewed as prescriptive for trend estimation in general, but were motivated by pragmatic considerations such as the current availability of only very short time-series of abundance estimates for some wide-ranging species; and the potential lack of uncertainty measures for baselines, especially if these are old. Hence, our results and recommendations apply in this narrow framing of data-poor situations (e.g., only a handful of point estimates are available), outside of which there are better alternatives. In particular, *Gerrodette (2011)* proposed a fully Bayesian approach to trend estimation, which may require the careful choice of adequate joint priors for the abundance estimates. Correct specification of the joint correlation structure may not be trivial, especially in long time series. More research needs to be carried out to recommend a default prior in this framework. Rather than using point abundance estimates (and their uncertainty), raw data (e.g. line transect data) should ideally be available to perform an integrated analysis of trend, along with abundance estimation (*Taylor et al., 2017*). This would allow the most flexibility to leverage all the information that may be available for a given species.

While we are aware that simple methods will necessarily ignore some of the complexities of data collected in specific and idiosyncratic contexts, we are nevertheless interested in the empirical performance of statistical methods in the spirit of evaluating their long-term frequency properties (Calibrated Bayes sensu *Little (2006)*, see also *Dorazio (2016)* for a pragmatic outlook in ecology and conservation). We think that our proposal, regularized regression with the weakly-informative prior of *Cook, Fúquene & Pericchi (2011)* offers a better alternative than the status quo. We are thereby challenging the current statistical norm in international conservation instruments such as the Marine Strategy Framework Directive and Habitats Directives in Europe. The challenge is not in the validity of the norm (but see *Amrhein, Greenland & McShane, 2019*), but in its application, because we think that the current stringent requirements may have rendered the legislation toothless if we have to wait for large and dramatic declines, associated with a higher risk of irreversible damage, to take actions. We showed that the status quo in trend analysis does not fare well with respect to the statistical properties invoked for its justification compared to our proposal. The latter is not a panacea though: it does not increase statistical power per se, but, within a context of nil null hypothesis testing, it should nevertheless be used for estimation and detection of trend with noisy and short time-series of abundances. The severe limitations on trend analysis with such frugal data underscore the need for (i) a re-alignment of current statistical practices with contemporary challenges in conservation; and (ii) for a more widespread and effectual application of the precautionary principle in conservation instruments.

## ACKNOWLEDGEMENTS

We thank our colleagues from the ICES Working Group on Marine Mammal Ecology (WGMME) and the OSPAR Marine Mammal Expert Group (OMMEG) for stimulating discussions. We thank Tim Gerrodette and Tomas Bird for critical and constructive comments.

### Funding
The authors received no funding for this work.

### Competing Interests
The authors declare that they have no competing interests.

### Author Contributions
- Matthieu Authier conceived and designed the experiments, performed the experiments, analyzed the data, prepared figures and/or tables, authored or reviewed drafts of the paper, and approved the final draft.
- Anders Galatius conceived and designed the experiments, authored or reviewed drafts of the paper, and approved the final draft.
- Anita Gilles conceived and designed the experiments, authored or reviewed drafts of the paper, and approved the final draft.
- Jérôme Spitz conceived and designed the experiments, authored or reviewed drafts of the paper, and approved the final draft.

### Data Availability
All R code and files to reproduce the analyses presented in the article are available in the Supplemental Files.

### Supplemental Information
Supplemental information for this article can be found online at http://dx.doi.org/10.7717/peerj.9436#supplemental-information.

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
