# Peer review of "Of power and despair in cetacean conservation: estimation and detection of trend in abundance with noisy and short time-series"

_PeerJ, doi:10.7717/peerj.9436_

## Round 0.1 · original submission · Minor Revisions

The points made during review are excellent - please carefully consider in your revision.

·

Basic reporting

The authors describe their approach clearly and demonstrate impressive background reading with their literature citations. The figures are well composed, and present a lot of information clearly and compactly. Their are a few minor issues in English. At line 126, for example, the plural of “scenario” in English is “scenarios” (despite its etymology!).

Experimental design

The authors motivate their simulations with a clear review of the difficulties of detecting trends in the populations of marine mammals. They state clearly how they hope their simulations will improve the situation. The paper would be strengthened if the authors connected their simulations to actual management goals. For example, effect size (r) ranged between 0.99 and 0.5. Is this range based on management goals within the EU framework? Would a decline of 40% (r=0.6) in, say, 10 years be acceptable? A technical point that I found puzzling was to generate the CVs from a uniform distribution (line 136). The precision of the estimates is related to the amount of sampling effort and to the technique (eg, line-transect vs mark-recapture), but it is under some control of the researchers, not a random variable.

Validity of the findings

Technically, the simulations were carried out as described. The results are clearly stated, and both positive and negative aspects of the results discussed.

The structure of the simulations, however, could affect the validity of the findings. The authors chose to simulate data and to carry out regressions in a way that did not conform to classical assumptions of linear regression. Because of this, they find that the classical (unregularized) approach without priors produces biased estimates of trend, inconsistent type 1 error rates, and inaccurate coverage. Their main conclusion that a weakly informative prior performs better than classical regression could be due to the structure of the simulations which caused the classical method to perform poorly. The authors should address this issue. If they believe that their non-standard regression procedures, including anchoring on the first point in the series, more accurately represents the way trend detection SHOULD be carried out with marine mammal data, they should explain why their non-standard approach is more appropriate. The regression methods they used would not be the methods used by most researchers.

Additional comments

I have 3 general comments.

First, I am uncomfortable with the authors' endorsement of a method (regression with a weakly informative prior) that will tend to produce underestimates of the size of a decline (lines 401-3). This is contrary to a precautionary approach, as the authors admit. They attempt to put a positive spin on the underestimation, but in my experience saying that a decline is “at least as large as x%” is not effective. The result will still be interpreted as a decline of x%. The recommended non-precautionary approach is inconsistent with the authors’ call “for a more widespread and effectual application of the precautionary principle in conservation instruments” (line 435).

Second, the authors base their estimates of a decline on an estimate of abundance at each time period. Thus, they consider that the information available (at each time) is a single number, the point estimate of abundance. However, a point estimate is a summary statistic for the available data at each time, and a lot of information is lost by using only this single number. Many papers have discussed the advantages of an integrated approach, which considers all the original data across all years within a single analysis. Gerrodette (2011) showed the power of this method specifically for the detection of a change in the vaquita population. The authors cited this paper, but did not implement or even discuss one of its main conclusions that would be relevant to their paper.

Third, “This paper hence deals with contesting a norm, not in challenging its validity but its current application” (line 383). Thus, the simulations were carried out within the flawed framework of null hypothesis significance testing, a method of inference which has been widely criticized, especially in recent years. Indeed, the authors present an excellent review of the issue. However, they have chosen not to challenge the null hypothesis paradigm in this paper. That is fine, but it means that the results of the paper will make, at most, an incremental improvement in the detection of trends in marine mammal populations, leaving all of us, as the title says, in a state of despair. I wish the authors had challenged the validity of the current norm. Perhaps they will in a future paper. That would be, in my opinion, a more substantial contribution.

·

Basic reporting

Basic reporting is well written. well cited, good structure. I have re-run much of the code and it seems well organised. I have not reproduced all results as the simulations take ~ 1 day each. However the authors did a really good job of their code and making it accessible.

Experimental design

A few comments on the text to improve clarity of the maths:

Data Assumptions:
In general please define parameters and subscripts before using them later on. For example t and r are both presented before their definitions.

Line 117: Please define the subscript ‘t’ early in the definitions section. I assume it refers to discrete time intervals t= 1…T. Also, it might be useful to define your temporal sampling scheme as having a total of T sampling occasions evenly spaced at times t=1…T. Perhaps defining the sampling scheme as the first assumption would clarify things.

Line 122, 123: I find the description of pt relative to r confusing. For instance in line 123, you express H0 as r=1. Strictly speaking, If r^[(t-1)/(T-1)] is exponentiated, then H0 is only true when both t and T are 1.

Line 123: This does not appear to be an assumption. I would specify it as a calculated parameter that arises from the assumptions.

Simulation Scenarii
Line 127: As above, Equation (1) does not seem to make sense. If r is the fraction of the original population remaining, then it should be expressed in terms of biomass or density.

Line 129: “… at time T > 1…” do you mean t>1?

Line 143: “We varied the value of r (= eb )” here you provide another definition for r. Please clarify this in the definitions and equations above.


Estimation
Line 173: “The location parameter of the Cauchy distribution”

Line 176: “of the unknown trend parameter r (b )” this appears to be another definition for r. Please make it obvious in the definitions if this is the case.


Case studies
Line 195: “We collected abundance or density estimates from published research or reports (Table 1).for density estimates from published research or reports (Table 1).” Please say how many.

Validity of the findings

Findings are robust and the application to real-world data is good and relevant.

Additional comments

Very nice work. So good to see a good simulation being used on real-world data.

---

## Round 0.2 · accepted · Accept

Congrats on a great contribution!